# Closed-loop recycling of sulfur-rich polymers with tunable properties spanning thermoplastics, elastomers, and vitrimers

Jin-Zhuo Zhao [1], Tian-Jun Yue [1], Bai-Hao Ren[1], Xiao-Bing Lu [1] & Wei-Min Ren [1]✉

The development of closed-loop recycling polymers that exhibit excellent performance is of great significance. Sulfur-rich polymers possessing excellent optical, thermal, and mechanical properties are promising candidates for chemical recycling but lack efficient synthetic strategies for achieving diverse structures. Herein, we report a universal synthetic strategy for producing polytrithiocarbonates, a class of sulfur-rich polymers, via the polycondensation of dithiols and dimethyl trithiocarbonate. This strategy has excellent compatibility with a wide range of monomers, including aliphatic, heteroatomic, and aromatic dithiols enabling the synthesis of polytrithiocarbonates with diverse structures. The present synthesis strategy offers a versatile platform for the construction of thermoplastics, elastomers, and vitrimers. Notably, these polytrithiocarbonates can be easily depolymerized via solvolysis into the corresponding monomers, which can be repolymerized to virgin polymers without changing the material properties.

The use of polymers brings great convenience to daily life owing to their low cost, lightweight, and wide range of multifunctionalities[1]. Over 350 million tons of polymers, of which most are polyolefins, are produced annually. However, their non-degradable features not only result in resource waste but also cause health and environmental issues with the widespread use of vast amounts of polymers[2,3]. The development of novel polymers capable of chemical recycling to monomers (CRM) is an effective strategy to address these issues. CRM transforms waste polymers directly back into monomers, which reduces the continuous need for monomers and the accumulation of waste plastics[4,5]. Consequently, an array of polymers capable of CRM has been developed using different recycling methods. For example, polyolefins containing fused rings or polyesters synthesized by the ring-opening polymerization (ROP) of lactones can be depolymerized into the corresponding monomers by heating the bulk materials[6–9]. Polycarbonates derived from the copolymerization of epoxides and $CO_2$ can also be recycled into the corresponding epoxides in the presence of catalysts[10–12]. Polyacetals with tensile strengths similar to those of commodity polyolefins and high thermal stabilities have been

demonstrated to be capable of CRM with a strong acid catalyst[13]. In addition, closed-loop recycling of polyethylene-like polycarbonates and polyesters has been achieved by the polycondensation of long-chain diols with carbonates or diesters and recycling via solvolysis[14]. In this context, achievements made in synthetic strategies have established a CRM platform that enables the straightforward construction of functional polymers with recyclability.

Sulfur-rich polymers, a category of functional polymers, have many attractive features, including a high refractive index[15–17], excellent adhesion to heavy metal ions[18–20], and electrochemical properties[21–23], owing to the incorporation of sulfur atoms into the polymer mainchain[24–26]. As a representative class of sulfur-rich polymers, polytrithiocarbonates exhibit enhanced properties owing to the high sulfur content of the thiocarbonate groups in the repeated units. In addition, polytrithiocarbonate-containing polymers exhibit anticancer and antimicrobial activities[27–29]. The copolymerization of episulfides with carbon disulfide ($CS_2$) constitutes a straightforward route for constructing polytrithiocarbonates[30,31]. However, the resultant polytrithiocarbonates are unrecyclable because of the high thermal

[1]State Key Laboratory of Fine Chemicals, Frontiers Science Center for Smart Materials, Dalian University of Technology, 2 Linggong Road, Dalian 116024, China. ✉e-mail: wmren@dlut.edu.cn

stability of the five-membered cyclic trithiocarbonates, which cannot be degraded to episulfides and $CS_2$ or converted into polytrithiocarbonates via ROP[32,33]. In contrast, the ROP of seven-membered cyclic trithiocarbonates enables the production of aliphatic polytrithiocarbonates. In particular, these polytrithiocarbonates exhibit superior optical, thermal, and mechanical properties compared to the corresponding polycarbonate analogs[32]. However, the limited number of monomers suitable for polytrithiocarbonate production significantly restricts the development of polymers with tunable properties. Therefore, the development of an efficient approach for the synthesis of recyclable polytrithiocarbonates with diverse structures remains challenging. In this study, we developed a universal synthetic methodology for producing polytrithiocarbonates that are viable for CRM through the polycondensation of dithiols with dimethyl thiocarbonate (DMTC). This strategy is highly compatible with a wide range of monomers, including aliphatic, heteroatomic, and aromatic dithiols. This compatibility enables the synthesis of polytrithiocarbonates with diverse structures, offering a versatile platform for the construction of polytrithiocarbonate-based materials such as thermoplastics, elastomers, and vitrimers (Fig. 1).

## Results

### Polycondensation of dithiols and DTMC

A previous study revealed that the construction of a trithiocarbonate group could be achieved via the coupling of thiols with thiophosgenes[34]. However, this process requires toxic compounds. We extended this work to construct polytrithiocarbonate via an alternative procedure wherein methyl trithiocarbonate (DMTC)[35] is utilized to substitute thiophosgenes for condensation polymerization with dithiols. The polycondensation procedure proceeded by heating the dithiol and DMTC mixture with the catalysis of alkali hydrides to accomplish the transesterification process, followed by the condensation of the polytrithiocarbonate oligomers generated in situ at elevated temperatures under reduced pressure to produce polytrithiocarbonates with high molecular weights (Fig. 2). Using 1,10-decanedithiol as the model monomer, the polycondensation conditions were screened with respect to the reaction temperature for the transesterification process and alkali metal hydride (Supplementary Table 1). The optimal procedure with the transesterification proceeding at 120 °C with potassium hydride and the polycondensation process proceeding at 180 °C afforded the highest efficiency and molecular weights. Using the optimized polycondensation procedure, various dithiols were used to construct polytrithiocarbonates with diverse structures. The C4–C18 aliphatic dithiols can be converted into the corresponding polytrithiocarbonates **P1** to **P7** with number-average molecular weights ($M_n$) ranging from 47.3 to 64.7 kg/mol (Fig. 2). The utilization of heteroatomic and aromatic dithiols allows for the incorporation of heteroatoms and aromatic rings into polytrithiocarbonate **P8** to **P11** with $M_n$s and Đ values ranging from 38.4 to 52.9 kg/mol and 2.02 to 2.18, respectively (Fig. 2). Consequently, a

series of polytrithiocarbonates with diverse structures were successfully synthesized in high yields in a facile and efficient manner.

Subsequently, the thermal, mechanical, and gas-barrier properties of the polytrithiocarbonates were investigated (Table 1). Polytrithiocarbonates **P1** to **P7** all exhibited a crystalline nature with melting ($T_m$) and crystalline ($T_c$) temperatures ranging from 92 to 106 °C and 69 to 88 °C, respectively, as characterized by the differential scanning calorimetry (DSC). (Table 1, entries 1–7, Fig. 3a and Supplementary Figs. 1–7). Notably, **P1** to **P3** possess higher $T_m$ than that of **P4** to **P7**, and the highest $T_m$ of 106 °C was observed from **P2**. The difference in the $T_m$s of these polytrithiocarbonates can be attributed to the different trithiocarbonate group density in the mainchain, as crystalline behavior of polymers is significantly affected by the configuration of methylene sequence in the repeat unit and the stacking ways of function groups[36,37]. Furthermore, the crystalline characteristics of **P1** to **P7** were characterized using powder X-ray diffraction (XRD) (Supplementary Fig. 8). As revealed, **P4** to **P7** possess a similar crystal structure, which are different from **P1** to **P3**. This can be attributed to the higher trithiocarbonate group density of **P1** to **P3** than that of **P4** to **P7**, resulting in the different stacking ways of polymer mainchain. The reason for the highest $T_m$ of **P2** may be attributed to its better crystal structure than other polytrithiocarbonates, as six diffraction peaks were observed for **P2**. In addition, no glass transition temperatures ($T_g$) were detected for these polytrithiocarbonates (Fig. 3a). This can be attributed to the high crystallinity nature of these polytrithiocarbonates (Supplementary Fig. 8). The high crystallinity of these polytrithiocarbonates indicates a few of amorphous phase domains existed in the aggregation state of these polytrithiocarbonates, which further results in the wide range of temperature of relaxation behaviors. The relaxation behavior of these polytrithiocarbonates can be observed from the tan δ curves in the DMA analysis, wherein the glass transitions of these polymers span a wide range of temperature (Supplementary Figs 15–21), suggesting longtime relaxation behaviors of these polytrithiocarbonates. In contrast, polytrithiocarbonates containing heteroatoms and aromatic rings (**P8** to **P11**) are amorphous with $T_g$ values ranging from −37 to 66 °C (Table 1, entries 8–11, Supplementary Figs. 9–12). In addition, all polytrithiocarbonates possess a 5% weight loss decomposition temperature ($T_{d5\%}$) of over 246 °C, as characterized by thermogravimetric analysis (Supplementary Figs. 13 and 14). Notably, the $T_{d5\%}$ increased with an increase in the length of methylene sequences in the dithiols from **P1** to **P7** (Supplementary Fig. 13). The mechanical performances of the polytrithiocarbonates were further examined via tensile testing using a universal testing system and dynamic mechanical analysis (Supplementary Figs. 15–23). According to the tensile tests, the Young's modulus ($E$), ultimate tensile strength ($\sigma_B$), and strain ($\varepsilon_B$) at break of **P1** reached 169 MPa, 27.7 MPa, and 316%, respectively. As the length of methylene sequence in the dithiols increased (**P1** to **P7**), the $\sigma_B$ values of the polytrithiocarbonates decreased and the $\varepsilon_B$ values increased (Fig. 3b). The $\sigma_B$ and $\varepsilon_B$ values of **P8** are much lower than those of **P3**, suggesting that the incorporation of heteroatoms into the polytrithiocarbonate mainchain weakens both the thermal and mechanical properties, thus leading to a soft material. In contrast, the incorporation of rigid aromatic rings into the polymer mainchain resulted in fragile **P10** with $E$, $\sigma_B$, and $\varepsilon_B$ values of 501 MPa, 20.7 MPa, and 9%, respectively (Table 1, entry 10). In particular, no mechanical performance was detected for **P9** and **P11** because of their sticky and brittle natures, respectively (Table 1, entries 9 and 11). Additionally, the storage moduli ($E'$) of **P1** to **P7** decreased from 2650 to 1292 MPa as the length of the methylene sequence in the dithiols increased (Fig. 3c). The $E'$ of **P8** and **P10** are 3423 and 3523 MPa, respectively. Moreover, the gas-barrier properties of the polytrithiocarbonates were tested using oxygen and water vapor transmission rate analyzers. The lowest oxygen transmission rate

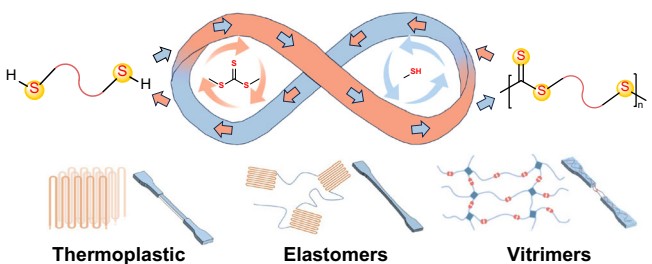

**Fig. 1 | Polycondensation of dithiols and dimethyl trithiocarbonate for synthesizing polytrithiocarbonates.** A general synthetic strategy for accessing diverse recyclable sulfur-rich polymers spanning thermoplastics, elastomers, and vitrimers.

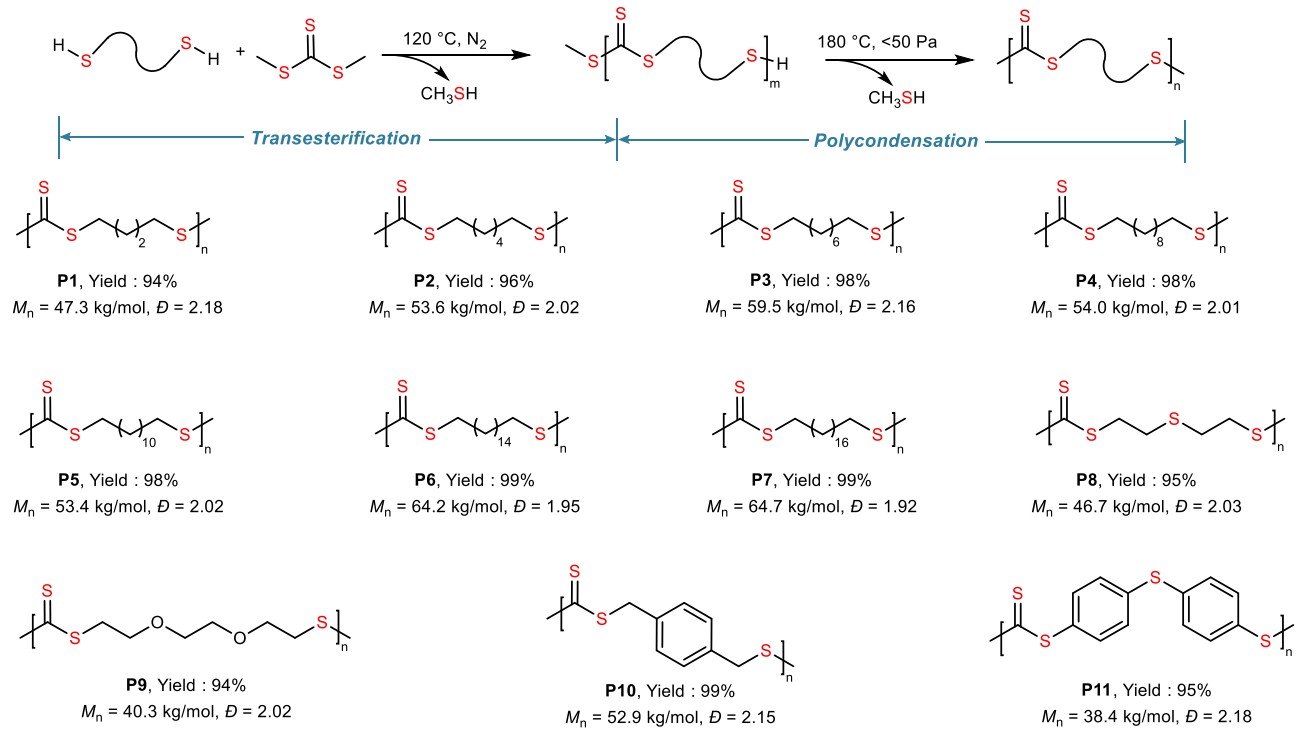

**Fig. 2 | Synthesis of a variety of polytrithiocarbonates from dithiols and DMTC.** The synthetic procedure contains transesterification and polycondensation processes.

**Table 1 | Properties of polytrithiocarbonates[a]**

| Entry | Polymer | $M_n^b$ (kg/mol) | $Đ^b$ | $T_g^c$ (°C) | $T_m^c$ (°C) | $T_c^c$ (°C) | $T_{d5\%}^d$ (°C) | $\sigma_B^e$ (MPa) | $\varepsilon_B^e$ % | $E^f$ (MPa) | $E'^g$ (MPa) | $O_2GTR^h$ cc/(m²·day) | MVTR[i] g/(m²·day) |
|---|---|---|---|---|---|---|---|---|---|---|---|---|---|
| 1 | P1 | 47.3 | 2.18 | n.d.[j] | 98.3 | 69.1 | 265 | 27.7 | 316 | 169 | 2650 | 1.62 | 0.59 |
| 2 | P2 | 53.6 | 2.02 | n.d. | 106.1 | 85.5 | 289 | 26.4 | 429 | 268 | 2600 | 18.00 | 2.41 |
| 3 | P3 | 59.5 | 2.16 | n.d. | 105.6 | 85.9 | 297 | 24.6 | 485 | 262 | 2174 | 50.28 | 2.32 |
| 4 | P4 | 54.0 | 2.01 | n.d. | 93.2 | 76.2 | 299 | 22.3 | 546 | 291 | 1880 | 59.37 | 1.81 |
| 5 | P5 | 53.4 | 2.02 | n.d. | 92.3 | 73.4 | 301 | 19.7 | 663 | 201 | 1662 | 108.46 | 2.11 |
| 6 | P6 | 64.2 | 1.95 | n.d. | 93.7 | 76.1 | 302 | 17.0 | 694 | 275 | 1593 | 131.20 | 2.98 |
| 7 | P7 | 64.7 | 1.92 | n.d. | 96.8 | 88.0 | 305 | 14.6 | 718 | 214 | 1292 | 136.27 | 1.66 |
| 8 | P8 | 46.7 | 2.03 | −18.7 | 56.2 | n.d. | 251 | 6.9 | 449 | 63 | 3423 | 4.29 | 1.06 |
| 9 | P9 | 40.3 | 2.02 | −37.2 | n.d. | n.d. | 292 | n.d. | n.d. | n.d. | n.d. | n.d. | n.d. |
| 10 | P10 | 52.9 | 2.15 | 14.4 | 114.3 | n.d. | 246 | 20.7 | 9 | 501 | 3523 | 6.06 | 1.75 |
| 11 | P11 | 38.4 | 2.18 | 66.1 | n.d. | n.d. | 258 | n.d. | n.d. | n.d. | n.d. | n.d. | n.d. |

[a]Polymerization performed at 120 °C and 70000 Pa for 2.0 h, followed by the polycondensation at 150 °C and 2000 Pa for 2.0 h and 180 °C and 50 Pa until 2.0 h after the Weissenberg phenomenon.
[b]The molecular weights and dispersities of the polymers were determined by gel permeation chromatography equipped with a triple detection array, including a differential refractive index (RI) detector, a two-angle light scattering (LS) detector, and a fourbridge capillary viscometer at 150 °C using 1,2,4-trichlorobenzene as the eluent.
[c]The glass transition temperature ($T_g$), melting temperature ($T_m$), and crystalline temperature ($T_c$) were determined using differential scanning calorimetry.
[d]Temperature at a molecular weight loss of 5% determined by thermogravimetric analysis.
[e]The breaking strength ($\sigma_B$) and elongation at break ($\varepsilon_B$) were determined by uniaxial tensile elongation testing.
[f]Young's modulus ($E$) calculated as the slope from 0% to 1% strain.
[g]Storage modulus at −40 °C determined by dynamic mechanical analysis.
[h]Oxygen permeation rate determined using an oxygen transmission rate analyzer.
[i]Water vapor transmittance determined using a water vapor transmission rate analyzer.
[j]Not determined.

($O_2GTR$) and water vapor transmission rate (MVTR) of 1.62 cc/(m²·day) and 0.59 g/(m²·day), respectively, were observed from **P1**. In contrast, the $O_2GTR$ and MVTR of poly(butylene carbonate) (PBC) are 53.04 cc/(m²·day) and 30.55 g/(m²·day)[38], respectively, and these values of high-density polyethylene (HDPE) are respective 157.78 cc/(m²·day) and 1.55 g/(m²·day)[39], suggesting that **P1** has much potential for use as gas-barrier materials. Interestingly, the $O_2GTR$ increased with an increase in the length of the methylene sequences in the dithiols. The superior gas-barrier properties of **P1** compared with those of other polytrithiocarbonates indicate its highly dense nature. Similarly, polytrithiocarbonates containing longer methylene sequences exhibit higher MVTRs. For **P8** and **P10**, the $O_2GTR$ and MVTR were less than 6.06 cc/(m²·day) and 1.75 g/(m²·day), respectively. The $O_2GTR$ and MVTR of **P9** and **P11** were inaccessible because they could not be processed into films using any available procedure.

## Synthesis of polytrithiocarbonate-based thermoplastic elastomer

With these diverse polytrithiocarbonates that exhibit distinct thermal and mechanical properties containing "hard" and "soft" materials in hand, the application of these polytrithiocarbonates to construct thermoplastic elastomers was explored. The "hard" **P3** oligomer and "soft" **P9** oligomer were used to construct multiblock copolymers with an alternating "hard" and "soft" structure (Fig. 4a). In particular, the adjustable feed ratio of dithiols **P3** oligomer to **P9** oligomer allows attaining **P12, P13,** and **P14** with a tunable "hard" to "soft" ratio, wherein the molar ratio of **3** to **9** in **P12, P13,** and **P14** are 3:1, 1:1, and 1:3, respectively (Supplementary Table 2). A melting peak at 79 °C, together with a crystalline peak at 53 °C, is observed for **P12**, and only a melting peak at 38 °C is observed for **P13**, as characterized by DSC (Supplementary Figs. 24 and 25). In contrast, only a glass transition peak at −46 °C is observed for **P14**, which presented as a

sticky polymer exhibiting no mechanical performance (Supplementary Fig. 26). In addition, mechanical property characterization revealed that **P12** possesses $\sigma_B$ and $\varepsilon_B$ values of 14.3 MPa 320%, respectively (Fig. 4b). The increase in the **P9** oligomer content in the multiblock copolymer results in the decrease in tensile strength but an increase in elongation of **P13** ($\sigma_B$ = 6.6 MPa, $\varepsilon_B$ = 332%) (Supplementary Table 2, entries 1 and 2). Additionally, the storage moduli ($E'$) at 25 °C of **P12** to **P13** decreased from 182 to 13 MPa as the content of **P9** oligomer increased from 25% to 50% (Supplementary Figs. 27 and 28). No yield points were detected for **P12** and **P13**, suggesting their elastomeric nature, which was further supported by the results obtained from cyclic tensile tests. As shown in Fig. 4c, hysteresis is observed for **P12** during the first cycle, together with a residual strain of 44% in the second cycle. In contrast, the hysteresis and residual strain (11%) of **P13** both decreased, suggesting a higher elasticity than **P12**. Notably, no obvious accumulation of permanent

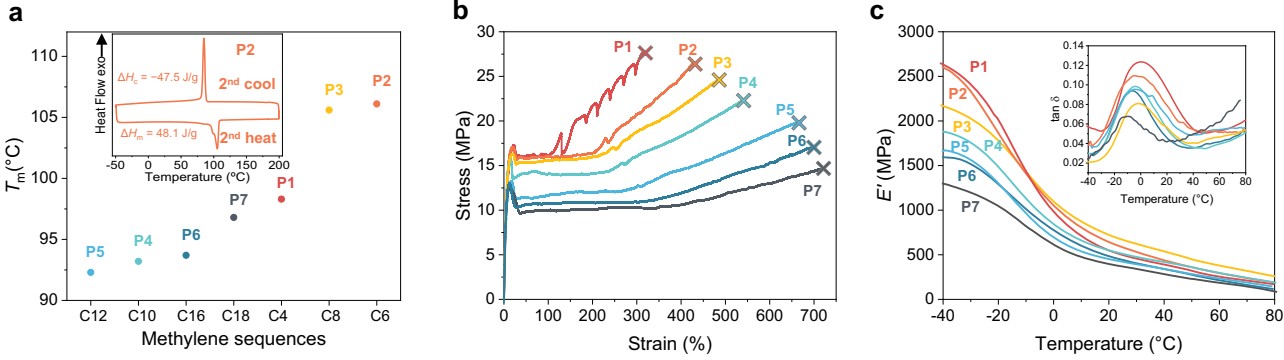

**Fig. 3 | Thermal and mechanical properties of P1 to P7. a** The $T_m$ values of the aliphatic polytrithiocarbonates. **b** Stress–strain curves. **c** Storage moduli.

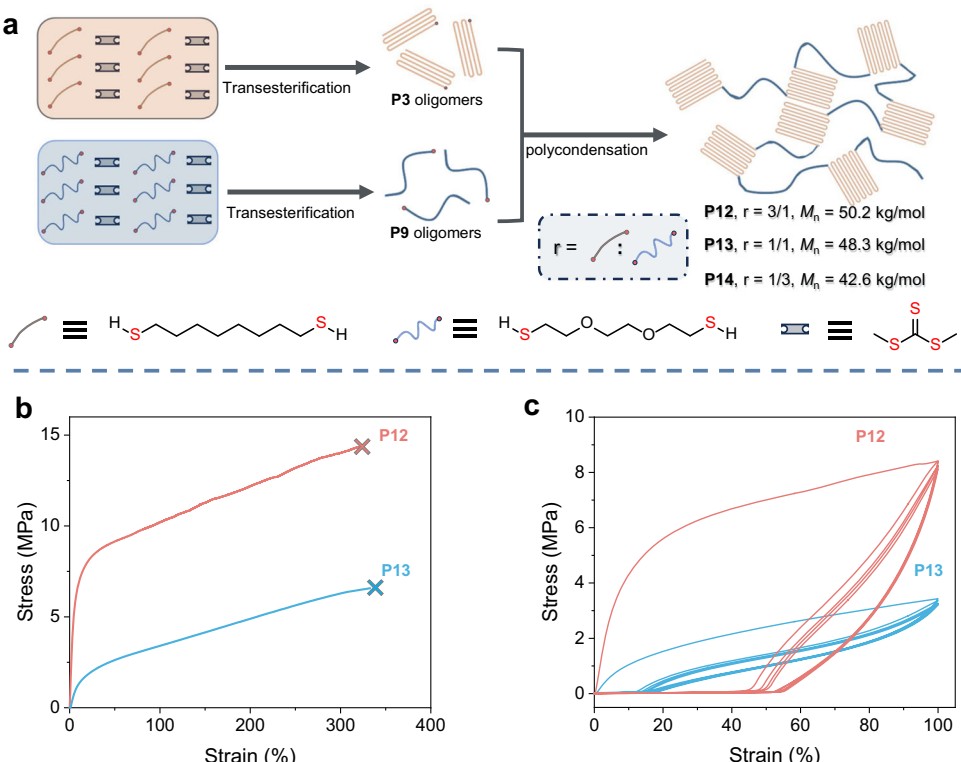

**Fig. 4 | Synthesis and characterization of polytrithiocarbonate-based elastomers. a** Schematic of the synthesis of copolymers **P12, P13,** and **P14. b** Stress–strain curves of **P12** and **P13. c** Cyclic tensile tests of the **P12** and **P13** for five cycles in

successive loading-unloading cycles at a strain of 100% under a deformation rate of 20 mm/min. **P14** is a sticky solid with a low $T_g$ of −46 °C and it is difficult to characterize its mechanical properties.

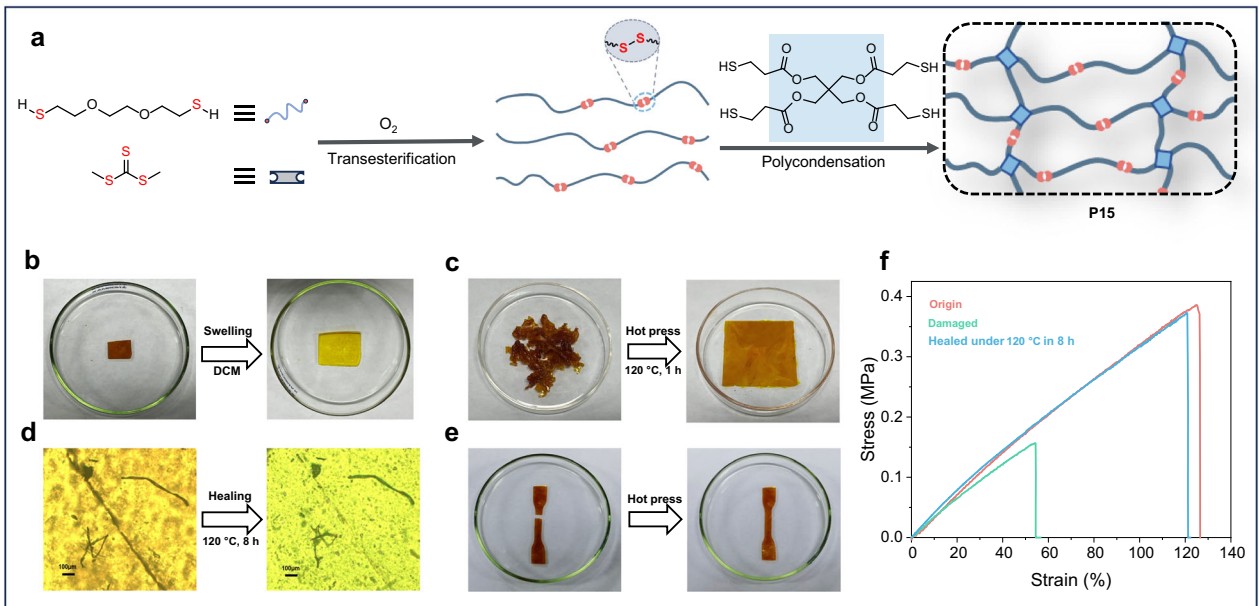

**Fig. 5 | Synthesis and characterization of polytrithiocarbonate-based vitrimer containing S–S bonds. a** Schematic of the synthesis of vitrimer. **b** Swelling test of the vitrimer in dichloromethane for 12 h. **c** The remodeling procedure of the vitrimer. **d** Optical microscopic images of the vitrimer before and after healing. **e** The healing procedure of the dumbbell-shaped vitrimer by hot pressing. **f** The stress–strain curves of the original, damaged, and healed vitrimer.

deformation was observed for these two elastomers, and almost no decay in the tensile strengths at the maximum strain point during continuous cycles was detected[40]. These results indicate the excellent fatigue resistance of these two elastomers, which was further supported by the rapidly diminishing and stabilized hysteresis within five cycles[41,42].

## Synthesis and characterization of polytrithiocarbonate-based vitrimer containing S–S bonds

The introduction of disulfide bonds, a typical dynamic covalent bond[43], into the polymer mainchain endows polymers with dynamic features, thus allowing them to be widely applied in materials science[44] as self-healing materials[45–47], drug delivery systems[48,49], and vitrimers[50–52]. In this study, the disulfide bond was expected to be formed via the oxidation of thiols by introducing a small amount of oxygen into the reaction system during the transesterification process. Consequently, an array of disulfide-containing polytrithiocarbonates derived from various dithiols was obtained (Supplementary Figs. 29–36). Notably, with dithiol **4** as the model monomer, the modulation of disulfide linkage content was achieved over a wide range from 1% to 18% by controlling the amount of air introduced (Supplementary Fig. 37). Subsequently, a vitrimer (**P15**) is envisioned to be formed by capitalizing on the polytrithiocarbonates derived from dithiol **9** with a disulfide linkage content of 18% to copolymerize with tetradentate thiol (Supplementary Fig. 38), that is pentaerythritol *tetrakis* (3-mercapto-propionate) (Fig. 5a). The swelling test of **P15** in dichloromethane indicated its crosslinked nature (Fig. 5b). In addition, the broken fragment of **P15** can be reprocessed by hot pressing at 120 °C under 10 MPa for 1 h (Fig. 5c). This can be attributed to exchange reactions between nearby disulfide bonds, which lead to network topology rearrangements. A dumbbell-shaped **P15** sample with a scratch was placed at 120 °C without pressure for 8 h. Subsequently, the scratch almost completely disappeared, as observed under an optical microscope (Fig. 5d). In addition, this vitrimer sample was capable of self-healing even when subjected to more serious damage (Fig. 5e). To evaluate the mechanical properties after self-healing, the mechanical properties of the original sample and damaged sample

before and after healing were further characterized using tensile tests. As shown in Fig. 5f, the damaged **P15** sample possesses a $\sigma_B$ and $\varepsilon_B$ of 0.16 MPa and 53%, respectively. In contrast, after self-healing, the damaged sample possesses comparable mechanical performances ($\sigma_B = 0.37$ MPa, $\varepsilon_B = 121\%$) to that of the original sample ($\sigma_B = 0.38$ MPa, $\varepsilon_B = 128\%$).

## Closed-loop recycling of polytrithiocarbonates

Recycling of polymers is an efficient strategy for reducing the negative effects of waste polymeric materials. Previous studies have demonstrated that sulfur-containing polymers[53–55], including polythioesters, polydisulfides, and polytrithiocarbonates, are viable for CRM. For the long methylene sequence-containing polytrithiocarbonates, the trithiocarbonate linkages in the mainchain can be readily broken using nucleophiles. In this context, long-chain polytrithiocarbonates are anticipated to be amenable to quantitative chemical recycling through reverse transesterification catalyzed by alkali hydroxides in methyl mercaptan (MeSH) by capitalizing on the in-chain functional groups as breakpoints (Fig. 6a).

For example, the yellow solid **P6** changed into red-orange after being treated with potassium hydroxide in MeSH at 50 °C for 24 h (Fig. 6b). Purified 1,16-hexadecanedithiol and DMTC were recovered in 97% yield after recrystallization. Notably, it was available to obtain **P6** directly from the mixture before separation through polycondensation manner. Many types of polymers are mixed during the practical sorting of mixed waste streams. Thus, the recycling of polytrithiocarbonates from polyolefin waste is desirable. To this end, pieces of commercial polypropylene (PP) (plastic solvent cap), HDPE (solvent bottle cap), and **P4** were subjected to the depolymerization procedure described above (Supplementary Fig. 39). Once the depolymerization procedure finished, PP and HDPE pieces were filtered out in their original unaltered states, whereas **P4** was completely depolymerized, affording 1,10-decanedithiol and DMTC in 96% yield. These results indicate that the obtained polytrithiocarbonates are viable for CRM via a reverse transesterification procedure and afford purified DMTC and dithiol monomers in high yields, thus establishing the closed-loop life cycles of polytrithiocarbonates.

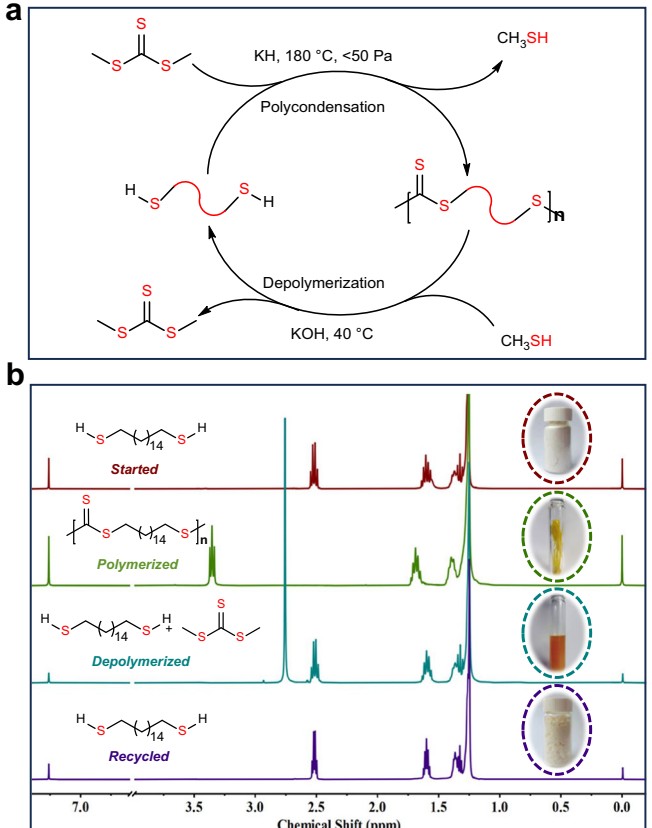

**Fig. 6 | Chemical recycling of polytrithiocarbonates. a** Schematic of the chemical recycling of polytrithiocarbonates. **b** Overlay of the ¹H NMR spectra in CDCl₃ of the closed-loop recycling of **P6**.

## Discussion

In summary, we developed an efficient strategy for chemically recyclable sulfur-rich polymers via the polycondensation of dithiols and DMTC. This method is compatible with a broad range of dithiols and produces polytrithiocarbonates with diverse structures. Additionally, this synthetic procedure enables the construction of polytrithiocarbonates with enriched architectures covering multiblock structures and dynamic networks, from which thermoplastic elastomers and vitrimers can be accessed using different dithiols. Moreover, the obtained polytrithiocarbonates were viable for CRM via reverse transesterification, thus endowing them with a promising application potential. Research on the application of polytrithiocarbonates in the fabrication of composites is ongoing.

## Method
### Materials

Iodomethane was purchased from Energy Chemical (Beijing, China), 1,4-butanedithiol, 1,6-hexanedithiol, 1,8-octanedithiol, 1,10-decanedithiol, 1,12-dibromododecane, 1,16-dibromohexadecane, bis(2-mercaptoethyl) sulfide, 1,4-benzenedimethanethiol and 4,4′-thiobisbenzenethiol were purchased from Aladin Reagent (Shanghai, China), 1,18-dibromooctadecane were purchased from Leyan Chemical (Shanghai, China). All other reagents for making monomers were purchased from Damao Chemical.

## Characterization methods
### NMR

¹H and ¹³C NMR spectra were recorded on a Varian INOVA-400 MHz type (¹H, 400 MHz; ¹³C, 100 MHz) spectrometer. Their peak frequencies were referenced versus an internal standard (TMS) shifts at 0 ppm for ¹H NMR and against the solvent, CDCl₃ at 77.4 ppm for ¹³C NMR, respectively.

### Gel permeation chromatography (GPC)

Molecular weight and molecular weight distribution of the polymers with low solubility at room temperature were determined by gel permeation chromatography (GPC) with the PL-GPC220 equipped with a triple detection array, including a differential refractive index (RI) detector, a two-angle light scattering (LS) detector, and a fourbridge capillary viscometer at 150 °C using 1,2,4-trichlorobenzene as the eluent.

### Differential scanning calorimetry (DSC)

The analysis of DSC was carried out with a METTLER TOLEDO DSC 3 thermal analyzer. Conditions: under $N_2$ atmosphere, the heating and cooling were at a rate of −10 and 10 K/min. Glass transition temperatures ($T_g$) were determined from the second heating cycle.

### Thermo-gravimetric analysis (TGA)

Thermo-gravimetric analyses of the resulted polymers were measured on a Mettler-Toledo TGA/SDTA851e.

### Dynamic mechanical analysis (DMA)

A rectangular spline for the Dynamic Mechanical Properties Test with a scale of 16 mm × 3.5 mm × 0.2 mm was obtained by cutting the film. The test was performed at small tension film mode, 0.1% strain, 1 Hz, 3.0 °C min⁻¹ on a Mettler-Toledo **DMA/SDTA 1+**.

### Uniaxial tensile elongation testing

The test was performed at a strain rate of 5 mm/min on an INSTRON 6800.

### Water vapor permeability

The moisture vapor transmission rate was determined at 37.8 °C with gas at 90% RH and 10 SCCM, following ASTM F1249 standard.

### Oxygen permeation analyzer

The oxygen transmission rate was determined at 23.0 °C under 100% oxygen concentration, following ASTM D3985 standard.

### Polarizing optical microscope (POM)

Optical microscope images were obtained on Zeiss Axioscope 5 polarizing microscope equipped with a Linkam hot stage.

### Wide angle X-ray diffraction (WAXD)

Powder X-ray diffraction data were collected on an EMPYREAN diffractometer with Cu KR radiation (λ = 1.54056 Å) over the 2θ range of 5–60 ° with a scan speed of 0.3333 °/s and a step size of 0.02 ° at room temperature. Crystallinity values of polymers were determined by deconvolution of WAXS diffractograms in MDI JADE 6.5 software.

## Data availability

The authors declare that the data supporting this study are available within the paper and the Supplementary information File. All other data is available from the authors upon request.

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

## Acknowledgements

W.-M.R. and T.-J.Y. acknowledge support from the National Natural Science Foundation of China (NSFC, 22171037 and 22101040). T.-J.Y. acknowledges the support from the China Postdoctoral Science Foundation (BX2021050 and 2021M690517). X.-B.L. acknowledges the support from the National Key Research and Development Program of China (2021YFA1501704). Gratitude is expressed to the Fundamental Research Funds for the Central Universities (DUT22LAB609).

## Author contributions

W.-M.R. conceived the project. W.-M.R. and T.-J.Y. directed the research. J.-Z.Z. carried out experiments and collected the overall data. J.-Z.Z and B.-H.R. analyzed data. J.-Z.Z and T.-J.Y. wrote the original manuscript draft, X.-B.Lu and W.-M.R. revised and finalized the manuscript. All co-authors discussed the results and commented on the manuscript.

## Competing interests

The authors declare no competing interests.
