## [Peer Review File · Nature Communications]

Closed-Loop Recycling of Sulfur-Rich Polymers with Tunable Properties Spanning Thermoplastics, Elastomers, and VitrimersReviewers' Comments:

Reviewer #1:

Remarks to the Author:

This manuscript, authored by Ren and co-workers, presents a significant advancement in the realm of recyclable sulfur-rich polymers. The noteworthy contribution lies in the approach of alkali hydrides-promoted polycondensation between dithiols and dimethyl thiocarbonate (DMTC), facilitating the efficient synthesis of high-molar-mass polytrithiocarbonates. Notably, this synthetic method allows for the production of sulfur-rich polymers featuring alkyl chains, heteroatoms, and aromatic groups, exhibiting exceptional thermal and mechanical properties. The manuscript also focuses on the divergent properties of thermoplastics, elastomers, and vitrimers. The vitrimers, synthesized with simplicity, exhibit remarkable self-healing properties, presenting a notable advantage over conventional polyesters. Additionally, the ability to recycle these polymers into monomers through solvolysis stands out as a significant feature, contributing to this manuscript's overall appeal. It is worth noting that the manuscript is well-referenced and supported, providing a strong foundation for the proposed advancements. So, I support publication of this work in Nature communication after minor revisions.

Suggested improvements:

1. A key focus in the revised version should emphasize the changes in the T_m across polymers P1 to P7, with special attention to highlighting the fact that polymer P2 demonstrates the highest T_m . An insightful discussion on the evolution of T_m as the polymer carbon chain grows should be integrated into the main text to enhance clarity and depth.
2. The authors investigated the barrier properties of the obtained polytrithiocarbonates, some of which performed much better than their carbonate analogues and HDPE. Further XRD measurements and crystallinity calculation will be helpful to explain the data and the trend among different polytrithiocarbonates.
3. For the synthesis of polytrithiocarbonate-based thermoplastic elastomer, the authors claimed P9 oligomer as the hard block. From P12 to P13 with an increase of P9 oligomer content, why did the polymers display a decreased stress and modulus (Table S2)? The increasing amount of hard block (P9) should theoretically make the polymer stiffer. The authors need to double check this section in the main text and Table S2, which seem not consistent with each other. For example, in the main text, it says the molar ratios of 3 to 9 in P12, P13, and P14 are 3:7, 5:5, and 7:3, respectively. But in Table S2, it shows the ratios are 3:1, 1:1, and 1:3.
4. In the context of monomer synthesis, it is recommended to explore the potential use of a sodium hydrosulfide/methanol system (2 h reflux) for directly converting the dibromide to dithiol. This alternative approach has the potential to significantly streamline the monomer synthesis process.
5. A few representative SEC traces of the synthesized polymers should be included in the SI.

Reviewer #2:

Remarks to the Author:

The submission of B-H Ren and coworkers on sulfur-rich polymers that can be recycled and that exhibit the properties of thermoplastics, elastomers or vitrimers depending upon the dithiol precursors used is an interesting piece of work, illustrating the case of polymers capable of chemical recycling to monomers.

It shows that polytrithiocarbonates of varying structures and with various mechanical properties can be obtained through transesterification and subsequent polycondensation from dithiols and methyl trithiocarbonate.

The synthetic approach leading to these polytrithiocarbonates is original, but the properties of the materials eventually obtained totally lack relevance.

For instance, the mechanical properties of polytrithiocarbonate thermoplastics are inferior to those exhibited by polyolefins such as HDPE or even LDPE. In their manuscripts (reference 13 and 14) both Coates and Mecking demonstrated that their polyacetals and polyethylene-like not only exhibit similar properties to those of HDPE and of LDPE but they can be recycled as well. They also manage to provide information about the renewable resources used to obtain their polyacetals and their polyethylene-like polycarbonates. Nothing comparable is discussed in the submission of B-H Ren, leaving the reader to wonder whether the monomers used are sourced from fossil or renewable feedstock.

Similar comments can be made on the properties of elastomers and vitrimers described in the submission of B-H Ren, where nothing quite remarkable is disclosed.

Overall, I recommend the manuscript of B-H Ren to be submitted to a more specialized journal.

Reviewer #3:

Remarks to the Author:

In this work, the authors described a new method for the synthesis of various chemical recyclable polytrithiocarbonates. This universal synthetic strategy allows for the construction of new materials covering thermoplastics, thermoplastic elastomers, and vitrimers. In general, the work is of high quality, giving a broad, complete, and correct overview of material properties and applications. I think the novelty is higher in the materials than in the method but overall high. For these reasons, I would recommend the work for publication in Nature Communications after resolving the amount of remarks suggested below.

1. In Figure 2, the authors show the yield of their polymers and the yield calculated from the weight of the precipitated polymer. However, the conversion of the monomers gives a much better idea about the efficiency of the polymerization reactions. I would recommend showing these values in Figure 2, instead of or complementing the polymer yields.

2. In Figure 2, the yield of the polymer decreases as the carbon number decreases in P1-P7. the yield of P1 is significantly lower than that of the other aliphatic carbonates, is that the lower the carbon number, the lower the reactivity? If so, what about the case of 1,2-ethanedithiol used in the polycondensation reaction?

3. As presented in Figure 6, many of the materials (P1-P7) had no detected glass transition, which should be further discussed.

4. For the copolymers of P3 and P9, more information should be given regarding the efficiency of the copolymerization reactions and the conversion of each oligomer.

5. The authors made efforts to synthesize thermoplastic elastomers by incorporating one soft and one hard homo-oligomer segment into multiblock copolymers. However, some of the homopolymers possess high strain, could these homopolymers also be called thermoplastic elastomers? In other words, is it a necessity to do these co-polymerizations?

6. In the synthesis of P15, an oligomer with 18% disulfide bond content was obtained after 1 L of oxygen was introduced. What is the disulfide bond content of the oligomer if no gas was bubbled in, or more oxygen was bubbled in?

Point-by-point response to the reviewers' comments

Reviewer 1

Question 1. *A key focus in the revised version should emphasize the changes in the T_m across polymers P1 to P7, with special attention to highlighting the fact that polymer P2 demonstrates the highest T_m . An insightful discussion on the evolution of T_m as the polymer carbon chain grows should be integrated into the main text to enhance clarity and depth.?*

Reply: Thanks a lot for the reviewer's kind suggestion. The crystalline behavior of polymers is complex, wherein the functional group and methylene sequence in the repeat unit may both affect melting temperature. Different methylene sequence lengths may result in different chain conformations and crystal structures when crystallizing (*Polymer* **2014**, 55, 1228–1248). Different chain conformations and crystal structures of the same substance result in widely varying melting temperatures (*Macromolecules* **2014**, 47, 236–245). Further characterizing **P1** to **P7** with XRD revealed that polytrithiocarbonates with different methylene sequence lengths have different crystalline structures (Figure R1). For **P4** to **P7**, they have similar crystals, and possessing similar T_{ms} (92 to 94 ° C). Whereas, the crystals of **P1** to **P3** are different to that of **P4** to **P7**. And the **P1** to **P3** possess higher T_{ms} than that of **P4** to **P7**. This can be attributed to the higher trithiocarbonate group density of **P1** to **P3** than that of **P4** to **P7**, resulting in the different stacking ways of polymer mainchain. Specifically, six peaks were observed at 2θ of 14.1° , 18.5° , 23.0° , 24.1° , 29.3° , 39.3° , and for **P2**. In contrast, only up to four diffraction peaks were observed for other polytrithiocarbonates. This result suggests better crystal structure of **P2**, thus possessing higher T_m , which can be attributed to the matched functional group interaction and methylene sequence configuration. Detail investigation on the crystalline behavior, as well as the influences on T_m of these polytrithiocarbonates are ongoing in our lab. Correspondingly, the description on accounting this result has been updated in the revised manuscript as: *“Notably, **P1** to **P3** possess higher T_m than that of **P4** to **P7**, and the highest T_m of 106 °C was observed from **P2**. The difference in the T_{ms} of these polytrithiocarbonates can be attributed to the different trithiocarbonate group density in the mainchain, as crystalline behavior of polymers is significantly affected by the configuration of methylene sequence in the repeat unit and the stacking ways of function groups.^{36,37} Furthermore, the crystalline characteristics of **P1** to **P7** were characterized using powder X-ray diffraction (XRD) (Supplementary Fig. 8). As revealed, **P4** to **P7** possess a similar crystal structure, which are different from **P1** to **P3**. This can be attributed to the higher trithiocarbonate group density of **P1** to **P3** than that of **P4** to **P7**, resulting in the different stacking ways of polymer mainchain. The reason for the highest T_m of **P2** may be attributed to its better crystal structure than other polytrithiocarbonates, as six diffraction peaks were observed for **P2**.”*

Figure R1. Powder XRD profiles of **P1** to **P7**.

Question 2. *The authors investigated the barrier properties of the obtained polytrithiocarbonates, some of which performed much better than their carbonate analogues and HDPE. Further XRD measurements and crystallinity calculation will be helpful to explain the data and the trend among different polytrithiocarbonates?*

Reply: Thanks for the reviewer's kind suggestion. According to the reviewer's suggestion, the crystalline characteristics of **P1** to **P7** were further explored using XRD. Correspondingly, the crystallinity of **P1** to **P7** were calculated in a range of 64% to 84% (Figure R1). They didn't show a tendency of linear increase or decrease with the increase of length of carbon chain and no matched relationship between the crystallinity and barrier properties can be concluded. Alternatively, the barrier properties of polymers can be significantly affected by the functional groups of the polymers. For instance, poly(ethylene terephthalate) (PET) which has lower value of crystallinity than HDPE, exhibits higher oxygen barrier performance and lower water barrier properties (*J. Drug. Deliv. Sci. Technol.* **2022**, *71*, 103330); poly(butylene carbonate) (PBC), which cannot undergo repeated crystallization, exhibits better oxygen barrier properties than highly crystalline HDPE. In this work, as the methylene sequences lengthen, the trithiocarbonate group content in the polytrithiocarbonates (**P1** to **P7**) has been changing, of which the oxygen transmission rate was increasing towards to that of HDPE. Particularly, Both polythiocarbonate and HDPE have good water vapor barrier properties, resulting in their ability to maintain good water vapor barrier properties without following a linear pattern.

Question 3. *For the synthesis of polytrithiocarbonate-based thermoplastic elastomer, the authors claimed P9 oligomer as the hard block. From P12 to P13 with an increase of P9 oligomer content, why did the polymers display a decreased stress and modulus (Table S2)? The increasing amount of hard block (P9) should theoretically make the polymer stiffer. The authors need to double check this section in the main text and Table S2, which seem not consistent with each other. For example, in the main text, it says the molar ratios of 3 to 9 in P12, P13, and P14 are 3:7, 5:5, and 7:3, respectively. But*

in Table S2, it shows the ratios are 3:1, 1:1, and 1:3.

Reply: Thanks for the reviewer's kind suggestion. After checking the experiment record, manuscript and supplementary materials, we found that the data in the SI table are correct. These typos errors were made during the drafting process and have been corrected in the revised manuscript.

Question 4. *In the context of monomer synthesis, it is recommended to explore the potential use of a sodium hydrosulfide/methanol system (2 h reflux) for directly converting the dibromide to dithiol. This alternative approach has the potential to significantly streamline the monomer synthesis process.*

Reply: Thanks a lot for the reviewer's kind suggestion. According to the reviewer's suggestion, we made an attempt to synthesize dithiol directly from 1,12-dibromododecane using a sodium hydrosulfide/methanol system. The conversion of the 1,12-dibromododecane reached 81% after refluxing for 3 hours under an inert atmosphere. Extending the reaction time to 10 hours resulted in complete conversion of the dibromide in another reaction (Figure R2). The dithiol was obtained with a yield of 69% after recrystallization. We appreciate the reviewer for suggesting a more convenient method for synthesizing the dithiol in our future work.

Figure R2. ^1H NMR spectra of: **a** the recrystallized 1,12-dodecanedithiol synthesized from 1,12-dibromododecane using hydrosulfide after refluxing 3 h; **b** the recrystallized 1,12-dodecanedithiol synthesized from 1,12-dibromododecane using hydrosulfide after refluxing 10 h.

Question 5. *A few representative SEC traces of the synthesized polymers should be included in the SI.*

Reply: Thanks for the reviewer's kind suggestion. SEC traces (Figure R3) of **P1** to **P7** have been presented in the revised supplementary materials as Supplementary Figs 73-79

Figure R3. GPC traces of (a) **P1**, (b) **P2**, (c) **P3**, (d) **P4**, (e) **P5**, (f) **P6**, (g) **P7**.

Reviewer 2

The synthetic approach leading to these polytrithiocarbonates is original, but the properties of the materials eventually obtained totally lack relevance. For instance, the mechanical properties of polytrithiocarbonate thermoplastics are inferior to those exhibited by polyolefins such as HDPE or even LDPE. In their manuscripts (reference 13 and 14) both Coates and Mecking demonstrated that their polyacetals and polyethylene-like not only exhibit similar properties to those of HDPE and of LDPE but they can be recycled as well. They also manage to provide information about the renewable resources used to obtain their polyacetals and their polyethylene-like polycarbonates. Nothing comparable is discussed in the submission of B-H Ren, leaving the reader to wonder whether the monomers used are sourced from fossil or renewable feedstock. Similar comments can be made on the properties of elastomers and vitrimers described in the submission of B-H Ren, where nothing quite remarkable is disclosed.

Reply: Thanks for the reviewer's suggestion. The development of polymers, especially those with high performances, that are viable for chemical recycling to monomers (CRM) is highly significant in terms of both environmental issues and resource reuse (*Nat. Chem.* **2016**, 8, 42; *Nature* **2021**, 590, 423; *Science* **2021**, 373, 783). Coates and Mecking have made good contributions for synthesizing chemical recyclable polymers, like polyacetal and polyethylene-like materials, respectively. And these polymers exhibit mechanical properties superior than that of HDPE. In our previous work, we have revealed that polytrithiocarbonate, derived from cyclic trithiocarbonate via ROP, exhibits superior performances to the corresponding polycarbonate, regarding thermal, mechanical, and optical performances, **suggesting that the incorporation of sulfur atoms into the repeat unit is of much value in improving the properties of polymers.** (*Macromolecules* **2022**, 55, 8651). However, limited by the structural diversity of cyclic trithiocarbonate, the obtained polytrithiocarbonate is weak in the structural diversity, which largely limits the

investigation of the performance-structural relationship of these polymers. **In response to this issue, we extend the synthetic method to a more general manner, that is, the polycondensation of dithiols and dimethyl thiocarbonate (DMTC). Benefited from the diversity of dithiols, various polytrithiocarbonates containing different methylene sequence in the repeat unit have been efficiently synthesized.** Correspondingly, the new polymers exhibited good thermal and mechanical properties. Notably, **P2** possesses a strength and strain at break of 26.4 MPa and 429%, which are much higher than that of the HDPE reported in reference 14 (Mecking's work) (Figure R4). Moreover, these polytrithiocarbonates possess high barrier properties, which is much higher than that of corresponding polycarbonates. These results revealed that the incorporation of sulfur atoms into polymer mainchain indeed does good to improving the performances of polymers.

On the basis of these results, the potential application of these polytrithiocarbonates were further explored by synthesizing polytrithiocarbonate-based thermoplastics, elastomers and vitrimers. The purpose for the properties and application exploration is to better present the full information of these new polytrithiocarbonates, which provides the fundament for the further investigation on this category of polymers.

Additionally, as presented in supplementary material, the dithiol is synthesized via the thiolation reaction of dibromoalkanes with thiourea. And the dibromoalkanes are synthesized from corresponding diols. Among these diols, 1,4-butanediol, 1,6-hexanediol, 1,10-decanediol, triethylene glycol, and 1,18-octadecanedithiol are derived from biomass, suggesting their renewable nature. Although other diols, like bis(2-mercaptoethyl) sulfide, 1,4-benzenedimethanethiol and 4,4'-thiobisbenzenethiol are not biomass-derived materials, the recyclable character of these polymers allows for their sustainable application. Moreover, the *reviewer 1* kindly suggested an alternative route for synthesizing dithiols from dibromoalkanes using a sodium hydrosulfide/methanol system, which makes it more facile to obtain these dithiols.

In fact, the precise synthesis of sulfur-containing polymers is of much challenge due to the intrinsic nature of sulfur, including strong coordinating ability and nucleophilicity. We also made great endeavor in the precise synthesis of sulfur-containing polymers, including main-chain structure control (*Angew. Chem. Int. Ed.* **2019**, *58*, 618; *Angew. Chem. Int. Ed.* **2022**, *61*, e202115950; *Macromolecules* **2022**, *55*, 8651–8658), stereochemistry control (*Angew. Chem. Int. Ed.* **2018**, *57*, 12670) as well as the topological structure exploration (*Angew. Chem. Int. Ed.* **2020**, *59*, 13633; *Angew. Chem. Int. Ed.* **2021**, *60*, 4315). This work presented here is another representative procedure for synthesizing polytrithiocarbonates. Hence, the work presented in this manuscript is of much significance in terms of whatever the efficient construction of chemical recyclable polytrithiocarbonates or acquisition of high performances. This study is aimed at providing a general synthetic method for

synthesizing polytrithiocarbonates, a type of sulfur-rich polymers, with diverse structures. The properties of these new polymers have been explored regarding thermal, mechanical, and barrier properties.

Figure R4. **a** Stress–strain curves of polytrithiocarbonates **P1** to **P7**, **b** Stress–strain curves of HDPE, PE-18,18 and PC-18 in Meching’s work.

Reviewer 3

Question 1. *In Figure 2, the authors show the yield of their polymers and the yield calculated from the weight of the precipitated polymer. However, the conversion of the monomers gives a much better idea about the efficiency of the polymerization reactions. I would recommend showing these values in Figure 2, instead of or complementing the polymer yields.?*

Reply: Thanks for the reviewer’s suggestion. The method employed for the polycondensation of dithiol and methyl trithiocarbonate (DMTC) is actually divided into two steps, that is a transesterification and polycondensation step. In this process, the polycondensation process didn’t started until the dithiol and DMTC were both fully transformed into polytrithiocarbonate oligomers. In this case, the conversion of each monomer is over 99% as monitored by the ^1H NMR spectroscopy analysis via interrupted sampling experiment. On the other hand, the total weight of the reaction mixture might be reduced due to the high reaction temperature and reduced pressure, which may cause the volatilization of small molecules by-products which could be pumped away. Therefore, the molecular weight, molecular weight distribution and yield of the polymer are more indicative of polycondensation efficiency than the conversion of monomers.

Question 2. *In Figure 2, the yield of the polymer decreases as the carbon number decreases in P1-P7. the yield of P1 is significantly lower than that of the other aliphatic carbonates, is that the lower the carbon number, the lower the reactivity? If so, what about the case of 1,2-ethanedithiol used in the polycondensation reaction?*

Reply: Thanks for the reviewer’s suggestion. The low yield of **P1** can be attributed to the generation of by product during the polycondensation rather than the low activity of dithiol. For the synthesis of **P1** via the polycondensation of 1,4-butanedithiol and DMTC, the seven-membered cyclic trithiocarbonate was detected as the byproduct during the transesterification process (Figure R5a), which caused the decrease in the yield of **P1**. In addition, according to the reviewer’s suggestion, the 1,2-ethanedithiol

was employed for producing polytrithiocarbonate via this method. However, a few of polytrithiocarbonates was obtained after the reaction finished. Further characterization of the reaction residue indicates the formation of five-membered cyclic trithiocarbonate (Figure R5b). This can be attributed to the higher thermodynamic stability of five-membered cyclic trithiocarbonate.

Figure R5. ^1H NMR spectra of: **a** the recrystallized 1,12-dodecanedithiol synthesized from 1,12-dibromododecane using hydrosulfide after refluxing 3 h; **b** the recrystallized 1,12-dodecanedithiol synthesized from 1,12-dibromododecane using hydrosulfide after refluxing 10 h.

Question 3. *As presented in Figure 6, many of the materials (P1-P7) had no detected glass transition, which should be further discussed.*

Reply: Thanks for the reviewer's kind suggestion. According to the reviewer's suggestion, the crystalline ability of **P1–P7** was further characterized by the powder X-ray diffraction (Figure R6). As revealed, all the polytrithiocarbonates possess a crystallinity of over 64%, indicating only a few of amorphous phase domains existed in the aggregation state of these polytrithiocarbonates, which results in an inapparent glass transition temperature at the macroscopic level. Correspondingly, the discussion on these results has been presented in the revised manuscript as: "*In addition, no glass transition temperatures (T_g) were detected for these polytrithiocarbonates (Fig. 3a). This can be attributed to the high crystallinity nature of these polytrithiocarbonates (Supplementary Fig. 8). The high crystallinity of these polytrithiocarbonates indicates a few of amorphous phase domains existed in the aggregation state of these polytrithiocarbonates), which further results in the wide range of temperature of relaxation behaviors.*"³⁸

Figure R6. Powder XRD profiles of **P1** to **P7**.

Question 4. *For the copolymers of P3 and P9, more information should be given regarding the efficiency of the copolymerization reactions and the conversion of each oligomer.*

Reply: Thanks for the reviewer's kind suggestion. The conversion of each oligomer can be determined by the ratio of the two components in **P12–P14**, which is the same as the molar ratio of the two reactants, as indicated by the ^1H NMR spectra in Supplementary Figs 70–72. As calculated, the yields of **P12–P14** (copolymers of **P3** and **P9**) were 98%, 98% and 96%, respectively. Correspondingly, these results have been updated in Supplementary Table 2 in the revised Supplementary materials.

Question 5. *The authors made efforts to synthesize thermoplastic elastomers by incorporating one soft and one hard homo-oligomer segment into multiblock copolymers. However, some of the homopolymers possess high strain, could these homopolymers also be called thermoplastic elastomers? In other words, is it a necessity to do these co-polymerizations?*

Reply: Thanks for the reviewer's kind suggestion. Indeed, some of the homopolymers exhibit high strain even to 700%. However, apparent yielding point was observed for each polytrithiocarbonate (**P1–P7**), indicating their plastic instead a thermoplastic elastomeric character. To better present the performances of the synthesized polytrithiocarbonates with diverse structures, we extended these polymers to thermoplastic elastomers and vitrimers, which indicates their promising application potential in various areas.

Question 6. *In the synthesis of P15, an oligomer with 18% disulfide bond content was obtained after 1 L of oxygen was introduced. What is the disulfide bond content of the oligomer if no gas was bubbled in, or more oxygen was bubbled in?*

Reply: Thanks for the reviewer's kind suggestion. According to the reviewer's suggestion, the Indeed, an oligomer with 10% disulfide bond content was obtained without bubbling oxygen and nitrogen, as shown in Supplementary Fig. 36. The content

of disulfide bond remained around 18% after the introduction of 2 L of oxygen. Similar results were obtained when 3 L of oxygen were added, attributed to the fact that the system's oxygen saturation was reached using only 1 L of oxygen before the addition of KH.

Table R1. The various contents of disulfide bonds in oligomer of **P15** with different volume of gas.

Gas	Volume of Gas (L)	Time (min)	Content of disulfide bonds (%)
N ₂	3.0	30	0
N ₂	2.0	20	1
N ₂	1.0	10	4
N ₂	0.5	5	8
none	0	0	10
O ₂	0.5	5	14
O ₂	1	10	18
O ₂	2	20	18
O ₂	3	30	18

Reviewers' Comments:

Reviewer #1:

Remarks to the Author:

The authors have adequately addressed my comments on the previous version of the manuscript.

Reviewer #3:

Remarks to the Author:

I think the authors have done an incredible job clarifying all reviewers' concerns and comments, I therefore support the publication of this manuscript in its current form.